# Quantifying Memorization Across Neural Language Models

**Nicholas Carlini**[*]    **Daphne Ippolito**[1,2]    **Matthew Jagielski**[1]
**Katherine Lee**[1,3]    **Florian Tramèr**[1]    **Chiyuan Zhang**[1]

[1]*Google Research*
[2]*University of Pennsylvania*
[3]*Cornell University*

## Abstract

Large language models (LMs) have been shown to memorize parts of their training data, and when prompted appropriately, they will emit the memorized training data verbatim. This is undesirable because memorization violates privacy (exposing user data), degrades utility (repeated easy-to-memorize text is often low quality), and hurts fairness (some texts are memorized over others).

We describe three log-linear relationships that quantify the degree to which LMs emit memorized training data. Memorization significantly grows as we increase (1) the capacity of a model, (2) the number of times an example has been duplicated, and (3) the number of tokens of context used to prompt the model. Surprisingly, we find the situation becomes more complicated when generalizing these results across model families. On the whole, we find that memorization in LMs is more prevalent than previously believed and will likely get worse as models continues to scale, at least without active mitigations.

## 1 Introduction

The performance of neural language models has continuously improved as these models have grown from millions to trillions of parameters (Fedus et al., 2021), with their training sets similarly growing from millions to trillions of tokens. In anticipation of future, even larger models trained on minimally curated datasets, it is important to quantify factors that lead to increased memorization of a model's training set. Indeed, recent work has shown that *training data extraction attacks* are a practical threat for current language models (Carlini et al., 2020); an adversary interacting with a pretrained model can extract individual sequences that were used to train the model.

While current attacks are effective, they only represent a lower bound on how much memorization occurs in existing models. For example, by querying the GPT-2 language model, Carlini et al. (2020) (manually) identified just 600 memorized training examples out of a 40GB training dataset. This attack establishes a (loose) lower bound that at least 0.00000015% of the dataset is memorized. In contrast, we are able to show that the 6 billion parameter GPT-J model (Black et al., 2021; Wang and Komatsuzaki, 2021) **memorizes at least 1% of its training dataset**: The Pile (Gao et al., 2020).

In addition to prior work's loose estimates of models' memorization capabilities, there is a limited understanding of how memorization varies across different neural language models and datasets of different scales. Prior studies of memorization in language models either focus on models or datasets of a fixed size (Carlini et al., 2019; Zhang et al., 2021; Thakkar et al., 2020) or identify a narrow memorization-versus-scale relationship (Carlini et al., 2020; Lee et al., 2021). While McCoy et al. (2021) broadly study the extent to which language models memorize, their focus is on how to avoid the problem and ensure novelty of model outputs, rather than on studying model risk through identifying the maximal amount of data memorization.

---

[*]Authors ordered alphabetically.

This paper addresses both of the above open questions by comprehensively quantifying memorization across three families of neural language models and their associated datasets. We leverage access to each model's original training set to provide order-of-magnitude more precise bounds on the amount of extractable data that an adversary could recover than in prior works.

We first construct a set of prompts from the model's training set. By feeding prefixes of these prompts into the trained model, we check whether the model has the ability to complete the rest of the example verbatim. This allows us to measure memorization across models, datasets, and prompts of varying sizes. We identify three properties that significantly impact memorization:

1. **Model scale:** Within a model family, larger models memorize 2-5$\times$ more than smaller models.
2. **Data duplication:** Examples repeated more often are more likely to be extractable.
3. **Context:** It is orders of magnitude easier to extract sequences when given a longer context.

Our analysis suggests that future research on neural language modeling will need to take steps to prevent future (larger) models from memorizing their training datasets.

## 2 RELATED WORK

There is extensive prior work that qualitatively studies memorization in neural language models. Prior work has demonstrated *extraction attacks* that recover memorized data including URLs, phone numbers, and other personal information (Carlini et al., 2020; Ziegler, 2021)—or synthetically injected "canaries" (Carlini et al., 2019; Henderson et al., 2018; Thakkar et al., 2020; Thomas et al., 2020). However most of these works are qualitative and aim to demonstrate the existence of extractable data, rather than precisely quantifying how much models memorize. For example, the unprompted memorization evaluation of Carlini et al. (2020) found just 600 examples of memorization in GPT-2. Our paper aims to establish tighter bounds on the fraction of a dataset that is memorized.

Our analysis is relevant to the broad literature on privacy attacks on machine learning. For example, membership inference attacks (Shokri et al., 2017; Yeom et al., 2018) let an adversary detect the presence of a given example in a model's training set; other forms of data leakage let an adversary learn dataset properties (Ganju et al., 2018; Fredrikson et al., 2015). We focus on extraction attacks due to their relevance for language modeling—extraction implies significant leakage from a model, and grows with data duplication (Lee et al., 2021), a common feature of large-scale text datasets.

Various definitions of memorization in deep neural networks have been studied in prior work (Carlini et al., 2019; 2020; Feldman and Zhang, 2020; Zhang et al., 2021). A detailed comparison with those existing formulations is presented in Section 3.1. One leading general memorization definition is differential privacy (Dwork et al., 2006), which formalizes the idea that removing any one example from the training set should not change the trained model. However, while differential privacy protects a single user's private information, it is ineffective for preventing memorization of highly duplicated data, and does not capture the complexity of social, linguistic data (Brown et al., 2022). Also, differentially private learning algorithms (Abadi et al., 2016) generally suffer from expensive computation, slow convergence, and poor model utility, despite recent advances (Anil et al., 2021).

In concurrent work, Kandpal et al. (2022) study how often models emit memorized data as a function of data duplication. Their analysis focuses on evaluating why training data extraction attacks succeed. In contrast, we explicitly prompt models with training data prefixes in order to measure memorization in the worst case, something that a practical attack cannot necessarily do.

**Prior scaling hypotheses.** Our motivation to study scaling phenomena stems from anecdotal evidence in prior work that memorization ability relates to various aspects of scale. In particular, our analysis on model scale is informed by preliminary experiments in (Zhang et al., 2017; Carlini et al., 2020), our data duplication experiments follow in the line of Lee et al. (2021), and our context length experiments build on hypotheses by Carlini et al. (2020); Ziegler (2021).

## 3 METHODOLOGY

### 3.1 DEFINITION OF MEMORIZATION

To begin, we first select a precise definition for memorization:

**Definition 3.1.** A string $s$ is *extractable with $k$ tokens of context* from a model $f$ if there exists a (length-$k$) string $p$, such that the concatenation $[p \| s]$ is contained in the training data for $f$, and $f$ produces $s$ when prompted with $p$ using greedy decoding.

For example, if a model's training dataset contains the sequence *"My phone number is 555-6789"*, and given the length $k = 4$ prefix *"My phone number is"*, the most likely output is *"555-6789"*, then this sequence is extractable (with 4 words of context). We focus on greedy sampling in this paper, and verify in Section 4.1 that our choice of decoding strategy does not significantly impact our results.

While prior work proposed other definitions, we prefer ours in this paper as it is more actionable. Some memorization definitions, including lower-bounds on differential privacy (Dwork et al., 2006; Jagielski et al., 2020; Nasr et al., 2021) or counterfactual memorization (Feldman and Zhang, 2020; Zhang et al., 2021), require training hundreds or thousands of models, which is impractical for large language models. Alternatively, computing *exposure* (Carlini et al., 2019) requires thousands of generations per sequence, and is only designed for carefully crafted training examples.Finally, $k$-eidetic memorization (Carlini et al., 2020), is a useful definition for *unprompted* memorization, but less useful for tightly bounding memorization by prompting with training data (as we will do). Future work might explore how our three scaling observations apply to other definitions of memorization.

## 3.2 Selection of Evaluation Data

Having chosen a definition, we next describe our evaluation procedure. Ideally, we would consider every sequence $x = [p \| s]$ in the model's training dataset (where $x$ has been split into a length-$k$ prefix $p$ and a suffix $s$). For each sequence, we would report if the model exactly reproduces $s$ when prompted with $p$, following Definition 3.1. Unfortunately, performing this test on every sequence in the training data would be prohibitively expensive. For example, the largest 6 billion parameter GPT-Neo model has a throughput of roughly one 100-token generation per second on a V100 GPU. Extrapolating to the 800GB training dataset, this would require over 30 GPU-years of compute.

Instead, we query on a smaller subset of the training data, that still produces statistically confident estimates. In this paper we randomly choose subsets of roughly 50,000 sequences, allowing us to efficiently run inference in just a few hours. The primary criteria when choosing a subset of the training data is to obtain a representative sample that allows us to draw meaningful conclusions from the data. We consider two approaches to constructing a subset of the data.

Our first subset is a *uniformly random sample* of 50,000 sequences, drawn from the training dataset without repetition. While a uniform sample is useful to estimate the absolute amount of memorization in a model, it is poorly suited for studying how memorization scales with data properties that are *not* uniformly represented in the training set. For example, prior work has identified that *data duplication* (i.e., how often the same sequence is repeated either exactly or approximately) is an important factor for memorization. Yet, because the frequency of training data duplication decays extremely quickly (Lee et al., 2021), a uniformly random sample of 50,000 sequences (accounting for $\leq 0.02\%$ of the dataset) is unlikely to contain *any* signal that would allow us to accurately measure the tail of this repeated data distribution. A similar concern arises for measuring how memorization scales with prompt length, since very long sentences account for only a small fraction of the training set.

Therefore, our second subset is a random sample *normalized* by both sequence lengths and duplication counts, which allows us to accurately measure memorization of large language models in the worst-case, on highly duplicated data with long prompts. For each sequence length $\ell \in \{50, 100, 150, \ldots, 500\}$, and integer $n$, we select 1,000 sequences of length $\ell$ that are contained in the training dataset between $2^{n/4}$ and $2^{(n+1)/4}$ times. We do this until we reach an $n$ for which 1,000 sequences are not available. This gives us 1,000 sequences that repeat between 6 and 8 times ($\approx 2^{11/4}$ and $\approx 2^{12/4}$) and also 1,000 sequences that repeat between 724 and 861 times ($\approx 2^{38/4}$ and $\approx 2^{39/4}$). This biased sampling allows us to more accurately measure memorization as a function of a sample's duplication factor and prompt length, without querying the entire dataset. Note that constructing this duplicate-normalized data subset requires some work, as efficiently identifying duplicate substrings in an 800GB training dataset is computationally challenging. We make use of the suffix array construction from Lee et al. (2021) (see Appendix).

For each length from 50 to 500 tokens, we collect 50,000 examples duplicated varying numbers of times, totaling roughly 500,000 sequences. For each sequence of length $\ell$, we prompt the model with

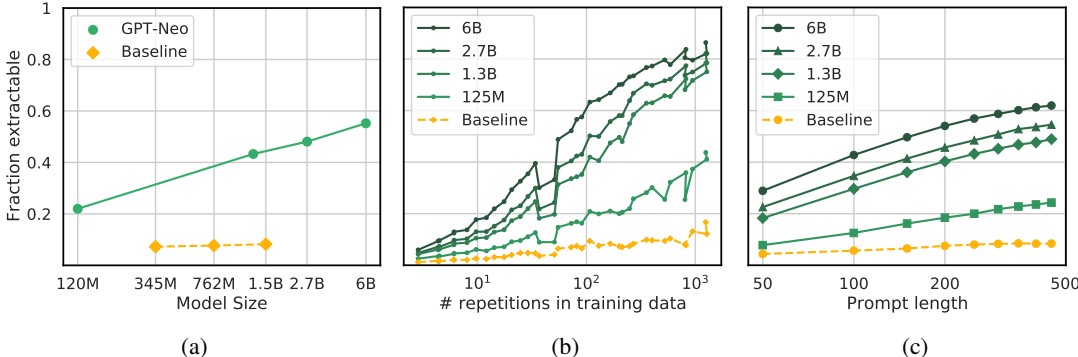

(a)  (b)  (c)

Figure 1: We prompt various sizes of GPT-Neo models (green) with data sampled from their training set—The Pile, and normalized by sequence lengths and duplication counts. As a baseline (yellow), we also prompt the GPT-2 family of models with the same Pile-derived prompts, even though these models were trained on WebText, a different training dataset. **(a)** Larger models memorize a larger fraction of their training dataset, following a log-linear relationship. This is not just a result of better generalization, as shown by the lack of growth for the GPT-2 baseline models. **(b)** Examples that are repeated more often in the training set are more likely to be extractable, again following a log-linear trend (baseline is GPT-2 XL). **(c)** As the number of tokens of context available increases, so does our ability to extract memorized text (baseine is GPT-2 XL).

the first $\ell - 50$ tokens and report the sequence as "extractable" if the model exactly emits the next 50 token suffix of this sequence. Fifty tokens corresponds to an average of 127 characters or 25 wordsin the GPT-Neo training set, well over the length of a typical English sentence. Finally, we compute the average probability that a sequence is extractable by averaging over all lengths $\ell$.

## 4 EXPERIMENTS

We primarily study the GPT-Neo model family (Black et al., 2021; Wang and Komatsuzaki, 2021) trained on the Pile dataset (Gao et al., 2020). The GPT-Neo models are causal language models trained with the objective of predicting the next token in a sequence given the previous ones. They come in four sizes: 125 million, 1.3 billion, 2.7 billion and 6 billion parameters.[1] The Pile is a dataset of 825GB of text collected from various sources (e.g., books, Web scrapes, open source code). Prior to the recent release of OPT (Zhang et al., 2022), the GPT-Neo models were the largest language models available for public download, and The Pile is the largest public text dataset available.

### 4.1 BIGGER MODELS MEMORIZE MORE

We begin by considering the impact of model size on memorization, expanding on prior studies which qualitatively established a relationship between the size of GPT-2 models and their ability to memorize <30 URLs (Carlini et al., 2020). In contrast, we study *a million* model generations in order to describe how model scale relates to memorization.

**Results.** We first study our biased random data sample normalized by duplication count and sequence lengths. The results of this experiment are given in Figure 1a. The y-axis reports the fraction of generations which exactly reproduce the true suffix for their prompt, averaged over all prompt and sequence lengths in our evaluation set. Because our biased sampling over-represents duplicated strings, the *absolute* degree of memorization in Figure 1a is not particularly important here—rather, we are interested in how memorization varies with scale.[2] We find that larger models memorize significantly more than smaller models do, with *a near-perfect log-linear fit* ($R^2$ of 99.8%): a ten fold increase in model size corresponds to an increase in memorization of 19 percentage points.

---

[1]As of February 2022, there is also a 20 billion parameter variant. Unfortunately this model uses a different training setup and tokenizer making it difficult to apply here.

[2]We repeat this experiment for a uniformly random subset of the data in Figure 2a.

To confirm that larger models are indeed *memorizing* more data, and not simply *generalizing* better, we repeat the analysis with the GPT-2 model family as a baseline. The GPT-2 models are similarly sized, and also trained on Internet-scraped data. If our "larger models memorize more" result was due to the predictive strength of larger models, and not the memorization of specific training data, we would expect a similar relationship between comparably sized GPT-2 models trained on similar data. Put differently, this baseline allows to establish what fraction of the training data is sufficiently "easy" that any language model can correctly predict the 50-token suffix, even if the example has not been seen during training. For example, a language model trained on multiple examples of number sequences can likely correctly complete some other unseen number sequences. We find that GPT-2 correctly completes approximately 6% of the examples in our evaluation set, compared to 40% for the similarly sized 1.3B parameter GPT-Neo model. A qualitative analysis (see examples in Appendix Figure 15) suggests that examples "memorized" by GPT-2 are largely uninteresting sequences (e.g., number sequences, repetitions of the same few tokens, or common phrases). Therefore, we conclude that larger models have a higher fraction of extractable training data because they have actually memorized the data; it is not simply that the larger models are more accurate.

## 4.2 REPEATED STRINGS ARE MEMORIZED MORE

Prior work provides preliminary evidence that memorization in language models increases with the number of times sequences are repeated in the training set (Carlini et al., 2020; Lee et al., 2021). We expand on this observation and quantitatively measure the effect of data duplication on memorization. Using our duplication-normalized data sample, we measure the fraction of sequences which are extractable, for buckets of sequences duplicated between 2 and 900 times. Each bucket consists of 1,000 distinct sentences, and we compute the average amount of memorization for each bucket.

**Results.** Figure 1b shows our results, aggregated over all sequence lengths. We observe a clear log-linear trend in memorization. While models rarely regurgitate strings that are repeated only a few times, this probability increases severely for highly duplicated strings. The small memorization values at low numbers of repetitions corroborates the positive impact of training dataset *deduplication* on memorization observed by Lee et al. (2021). However, we find that memorization does still happen, even with just a few duplicates—thus, deduplication will not perfectly prevent leakage. While this relationship is perhaps obvious, and has been corroborated for specific training examples in prior work (Carlini et al., 2019; 2020), our results show that it holds *across the entire training set*.

## 4.3 LONGER CONTEXT DISCOVERS MORE MEMORIZATION

The previous two questions evaluated how data collection and model training decisions impact the leakage of a model's training data when it is provided a fixed number of tokens from a sequence as context. As a result, those experiments suggest particular actions that could be taken to mitigate memorization (by reducing model size, or limiting the number of duplicate examples).

However, even when the model is fixed, it is possible to vary the amount of extractable training data by controlling the length of the prefix passed to the model. By studying how the number of tokens of context impacts extractability, we demonstrate the difficulty of *discovering* memorization—language models may only exhibit their memorization under favorable conditions.

**Results.** In Figure 1c, we observe that the fraction of extractable sequences increases log-linearly with the number of tokens of context. For example, 33% of training sequences in our evaluation set are extractable from the 6B model at 50 tokens of context, compared to 65% with 450 tokens of context. We call this the **discoverability phenomenon**: some memorization only becomes apparent under certain conditions, such as when the model is prompted with a sufficiently long context.

The discoverability phenomenon may seem natural: conditioning a model on 100 tokens of context is more specific than conditioning the model on 50 tokens of context, and it is natural that the model would estimate the probability of the training data as higher in this situation. However, the result is that some strings are "hidden" in the model and require more knowledge than others to be extractable.

From one point of view, it is good that some memorization is difficult to discover. This makes it harder for attackers to perform training data extraction attacks (Carlini et al., 2020), or otherwise exploit memorization. Indeed, if an exact 100 token prompt is required to make the model output a given string, then, in practice, an adversary will likely be unable to perform the attack. The difficulty

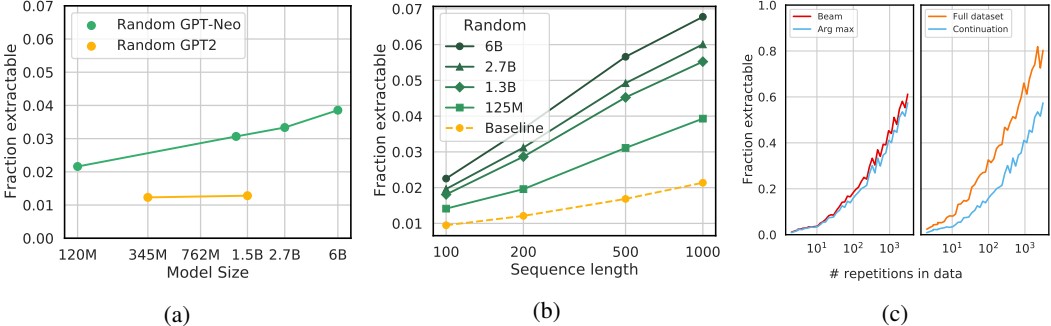

Figure 2: **(a)** Fraction of sequences extracted as a function of model scale where we sample uniformly from the training set. **(b)** Fraction of sequences extracted as we vary the length of the prompt. For each sequence length $n$, $n$-50 tokens are used as the prefix, and we check for extraction of the remaining 50 tokens. **(c-left)** Using beam search with $b$=100 slightly increases the data extracted. **(c-right)** We observe considerably more memorization when checking whether the generated sequence occurs anywhere in the entire training set (Section C). However, this approach is very computationally expensive so we do not use it for our other experiments.

in discovering memorization also reduces the likelihood of *non-adversarial* training data regurgitation. For example, the GitHub Copilot model (Chen et al., 2021) reportedly rarely emits memorized code in benign situations, and most memorization occurs only when the model has been prompted with long code excerpts that are very similar to the training data (Ziegler, 2021). Practitioners building language generation APIs could (until stronger attacks are developed) significantly reduce extraction risk by restricting the maximum prompt length available to users.

Viewed differently, however, the difficulty of discovering memorization can also harm our ability to audit privacy in machine learning models. Because provably-correct approaches for privacy-preserving training of machine learning models are applied only rarely in practice (Abadi et al., 2016; Thakkar et al., 2020; Ramaswamy et al., 2020), it is common to attempt post-hoc *privacy auditing* (Jayaraman and Evans, 2019; Jagielski et al., 2020; Nasr et al., 2021). Our results suggest that correctly auditing large language models likely requires prompting the model with training data, as there are no known techniques to identify the tail of memorized data without conditioning the model with a large context. Improving upon this limitation is an interesting problem for future work.

### 4.4 ALTERNATE EXPERIMENTAL SETTINGS

In this section, we briefly review other strategies that we could have used to quantify memorization.

**Random dataset sampling.** The majority of this paper uses subsets of the training data that were explicitly sampled according to training data duplication frequency. Now, we consider how our results would differ if we chose a truly random subset of the training data, where each sequence is sampled uniformly, instead of sampling a duplicate-normalized dataset. Specifically, we randomly sample 100,000 sequences of varying lengths from The Pile dataset, then prompt the model and test for memorization as before (more details in Appendix C).

Figure 2a and Figure 2b present the results. We observe similar qualitative trends with model scale and context length as in Figure 1. Larger models memorize more training examples than smaller models—and much more than the GPT-2 models that were not trained on The Pile. Similarly, providing more context to a model increases the likelihood we discover memorization. We can extract the last 50 tokens of a length-1000 sequence with 7% probability for the largest GPT-J 6B model, compared to 4% probability for the smallest 125M GPT-Neo model. (And both of these are much larger than the 2% probability of extraction for the 1.5B parameter GPT2-XL model.) These results, taken together, allow us to estimate a lower bound that there is at least 1% of The Pile dataset that is extractable by the 6B GPT-J model, but not by GPT-2 XL.

**Alternate decoding strategies.** We have defined memorization as a model's ability to generate the true continuation when choosing the *most likely* token at every step of decoding. Yet, this greedy decoding strategy does not produce the overall most likely sequence. Many language model

| Prompt | Continuation (== 6B) | 2.7B | 1.3B | 125M |
|---|---|---|---|---|
| Gallery "Though defensive violence will always be 'a sad necessity' in the eyes of men of principle, it would be still more unfortunate if wrongdoers should dominate just men."- St. Augustine "A new idea is first | condemned as ridiculous, and then dismissed as trivial, until finally it becomes what everybody knows." - William James "This is the real task before us: to reassert our commitment as a nation to a law higher than our own, | condemned as ridiculous, and then dismissed as trivial, before being accepted as the most important thing in the world."- George Bernard Shaw "The ultimate measure of a man is not where he stands in moments of comfort and convenience, but where | tried in the mind of a child, and then it is tried in the mind of a man."- Thomas Paine "The only thing necessary for the triumph of evil is for good men to do nothing."- Edmund Burke " | and foremost a man's duty to his fellow man. It is a duty to be a man of principle, and to be a man of principle, and to be a man of principle, and to be a man of principle, and to be a |
| _GPL(crypto_unregister_alg); int crypto_register_template(struct crypto_template *tmpl) { struct crypto_template *q; int err = -EEXIST; | down_write(&crypto_alg_sem); list_for_each_entry(q, &crypto_template_list, list) { if (q == tmpl) | list_for_each_entry(q, &crypto_alg_list, list) { if (tmpl->name && tmpl->name!= q->alg.cra_name) | q = kzalloc(sizeof(*q), GFP_KERNEL); if (!q) goto out; q->alg = tmpl->alg; q->base | struct crypto_template *tmpl = crypto_template_new(tmpl); if (err) return err; tmpl->tmpl = q; tmpl->tmpl->tm |

Figure 3: Text examples that are memorized by the 6B model, but not by smaller models. Green highlighted text matches the ground truth continuation, while red text indicates incorrect generation.

applications use other decoding strategies, such as beam search to find the generation with highest likelihood. To understand how our choice of decoding strategy affects the amount of memorization we measure, we compare greedy decoding with beam search in Figure 2(c). We find that using beam search with 100 beams results in marginally more extracted memorization. The difference in extractable memorization is just under 2 percentage points on average, with a maximum of 5.6%. Interestingly, beam search and greedy decoding generated the same output 45% of the time.

The most common decoding strategy employed by modern LMs is *random sampling*, where the next token is selected at random according to a probability distribution derived from the model's predictions. McCoy et al. (2021) found that random sampling resulted in generated text with a greater number of novel $n$-grams. Since the goal of our study is to maximize discoverability—an antithetical goal to maximizing linguistic novelty—we do not present experiments that use random sampling.

**Alternate definition of extractability.** Our main experiments report a sequence as "extractable" if the model's generation is identical to the true suffix of the considered training example. However it is possible this suffix is still present (elsewhere) in the dataset. We now consider a loose lower bound on memorization that considers a sequence memorized if the generation $[p||f(p)]$ from a prompt $p$ is contained *anywhere* in the training dataset. Searching within the entire dataset finds more memorized content than comparing with the ground truth (Figure 2c). For examples at 100 repetitions, 32.6% of outputs are contained somewhere in the dataset but just 15.8% match the ground truth continuation.

## 4.5 QUALITATIVE EXAMPLES OF MEMORIZATION

In Figure 3, we present qualitative examples that are only memorized by the largest (6B) model, but not the smaller ones. We highlight some interesting patterns in these sequences: while the generations from the smaller models do not match the training data, they are generally thematically-relevant and locally consistent. However, a closer inspection reveals that those generations are only syntactically sound, but semantically incorrect. Appendix Figure 8 shows further examples of sequences that are memorized by *all* the models. We found most of these universally-memorized sequences to be "unconventional" texts such as code snippets or highly duplicated texts such as open source licenses. Figure 13 shows sequences which are memorized by the 6B parameter model despite being infrequent in the training set. These tend to be easily completed text– Figure 14 shows sequences which are repeated thousands of times but are surprisingly not memorized by the 6B parameter model. Many of these are mostly correctly completed, only differing on semantically unimportant characters.

## 5 REPLICATION STUDY

The above analysis provides evidence that memorization scales log-linearly with model size, data duplicates, and context length. We now replicate this analysis for other language models trained with different datasets and training objectives, namely: (1) the T5 family of models trained on the C4 dataset (Raffel et al., 2020), (2) models from Lee et al. (2021), trained on a deduplicated version of C4, and (3) the OPT family of models (Zhang et al., 2022), also trained on the Pile. We expected our results to cleanly generalize across settings, and this is indeed true for model scale. Yet, the situation is more complicated when considering data duplication, due to training set idiosyncrasies.

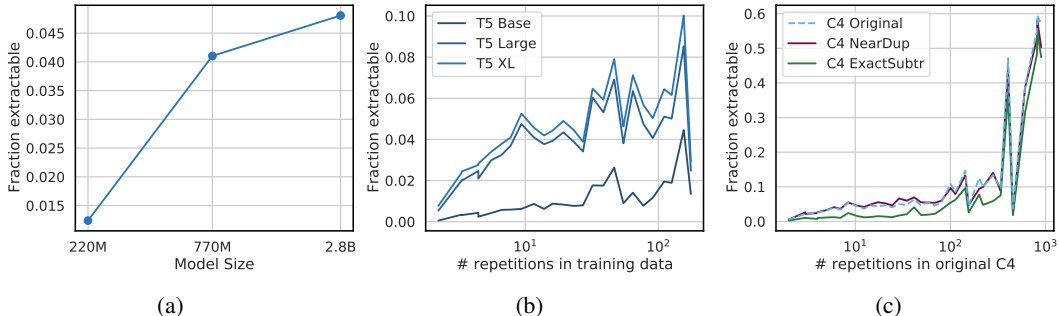

Figure 4: **(a)** Masked language model objective: Larger models have a higher fraction of sequences extractable on T5. **(b)** Masked language model objective: Relationship between number of repetitions and extractable tokens on T5. **(c)** Causal language model objective: Relationship between number of repetitions and memorization on language models trained with deduplicated data.

## 5.1 T5 MASKED LANGUAGE MODELING

**Model and dataset.** The T5 v1.1 models are masked encoder-decoder models trained to reproduce randomly deleted spans from an input sequence. The models vary in size from 77M to 11B billion parameters, and are trained on C4—a 806 GB curated version of English web pages from the Common Crawl. The largest T5 model (11B parameters) is the largest publicly available masked language model. T5 models are thus good candidates for studying how memorization scales with model size.

We must first define what is meant by "extractable data" for the masked language modeling task. T5 models are trained by removing a random $15\%$ of tokens from each training sequence (i.i.d), and the model must then "fill in the blanks" to restore the tokens that were dropped from the input. As a result of this different training objective, Definition 3.1 is not directly applicable: the model does not operate on a *prefix* and output a *suffix*. We instead call a sequence memorized if the model *perfectly* solves the masked language modeling task on that sequence. For example, we call a 200-token sequence memorized if the model can use the $170 (= 200 \cdot 0.85)$) tokens of context to perfectly predict the remaining 30 tokens $(= 200 \cdot 0.15)$. Because this token-dropping procedure is stochastic, it is possible that one set of dropped tokens might yield an output of "memorized" and another might not. For simplicity, we inspect only one set of masked tokens per sequence; because we are already averaging over 50,000 sequences this additional randomness does not harm the results of our analysis.

**Results.** In Figure 4a, we reproduce the model scaling effect (from Figure 1a) for T5 models. Larger models similarly have an increased ability to perfectly solve the masked prediction task. Surprisingly, while a scaling trend does hold here as well, the absolute memorization in masked models is an order of magnitude lower than for comparably sized causal language models. For example, the 3B parameter T5-XL model memorizes $3.5\%$ of sequences repeated 100 times, whereas the GPT-Neo 2.7B model memorizes $53.6\%$ of sequences repeated 100 times (with 150 tokens of context).

Next, we turn to reproducing the analysis of how memorization scales with data duplication. The situation here becomes significantly less clear. As shown in Figure 4b, sequences duplicated more often tend to be easier to memorize, but there is no monotonic scaling relationship. Compared to the case of the GPT-Neo models trained on The Pile, the relation between data duplication counts and memorization for T5 models trained on C4 exhibits large variance. This variance is *statistically significant*: sequences repeated 159 to 196 times are memorized with probability less than $5.1\%$ with $99.7\%$ confidence (three standard deviations from the mean), however sequences repeated 138 to 158 times (that is, *less often*) are memorized with probability at least $6.2\%$ (also with $99.7\%$ confidence). That is, for some reason, sequences that occur $\sim$140 times are *more likely to be memorized, despite occurring less often*, even if we assume a three-sigma error in both measurements simultaneously.

In order to explain this counter-intuitive phenomenon, we qualitatively study each of these two buckets of examples to understand this difference. We find that most of the duplicate examples repeated 138-158 times consist mainly of whitespace tokens. These sequences are thus much easier to predict correctly than other sequences, even if they are repeated more often. This effect, to a lesser extent, can be found in other buckets which contain many approximately near duplicates.

## 5.2 LANGUAGE MODELS TRAINED ON DEDUPLICATED DATA

**Model and dataset.** The models used in Lee et al. (2021) are 1.5B parameter causal language models. This model family consists of one model trained on C4 (the same dataset as T5), one model trained on a version of C4 that was deduplicated by removing all documents which were near-duplicates of other documents, and one model trained on a version of C4 that was deduplicated by deleting any string of length-50 tokens that occurred more than once. Lee et al. (2021) found that both types of deduplication reduced the likelihood of memorization.

**Results.** We were most interested in whether models trained on deduplicated data would still exhibit increased memorization of examples which were repeated frequently in the original, non-deduplicated C4 dataset (e.g., because the deduplication missed some near-duplicates). Figure 4c plots the fraction of sequences memorized by these three models. We draw two interesting conclusions from this data.

First, we confirm that models trained on deduplicated datasets memorize less data than models trained without deduplication. For example, for sequences repeated below 35 times, the exact deduplicated model memorizes an average of 1.2% of sequences, compared to 3.6% without deduplication, a statistically significant ($p < 10^{-15}$) decrease by a factor of $3\times$. Second, while deduplication does help for sequences repeated up to ∼100 times, it does not help for sequences repeated more often! The extractability of examples repeated at least 408 times is statistically significantly higher than any other number of repeats before this. We hypothesize that this is due to the fact that any deduplication strategy is necessarily imperfect in order to efficiently scale to hundreds of gigabytes of training data. Thus, while it may be possible to remove *most* instances of duplicate data, different and valid definitions of duplicates can mean deduplication is not exhaustive.

## 5.3 LANGUAGE MODELS TRAINED ON A MODIFIED VERSION OF THE PILE

**Model and dataset.** We finally study the OPT family of models (Zhang et al., 2022), that vary from 125 million to 175 billion parameters.[3] These models were trained on a 800GB dataset that overlaps with The Pile but is not identical and contains data from many new sources, while also removing some data from the Pile. This dataset was also deduplicated prior to training, and so we do not expect to see duplicate sequences memorized (much) more than sequences repeated only a few times.

**Results.** Overall, we find that while there are nearly identical scaling trends to those we found for GPT-Neo's model family, the effect size is orders-of-magnitude smaller (figure 7). Even the 66 billion parameter model memorizes a smaller fraction of The Pile than the smallest 125 million parameter GPT Neo model. This suggests two possible conclusions: (a) careful data curation and training can mitigate memorization, or (b) even slight shifts in data distribution can significantly alter what content gets memorized. Without direct access to the original training dataset, we can not distinguish between these two conclusions and hope future work will be able to resolve this question.

## 6 CONCLUSION

Our paper presents the first comprehensive quantitative analysis of memorization in large language models, by re-processing the training set to find memorized data. Our work has two broad conclusions.

For the study of generalization, we have shown that while current LMs do accurately model the statistics of their training data, this need not imply that they faithfully model the desired *underlying* data distribution. In particular, when the training data distribution is skewed (e.g., by containing many duplicates of some sequences) larger models are likely to learn these unintended dataset peculiarities. It is therefore important to carefully analyze the datasets used to train ever larger models, as future (larger) models are likely to remember even more training details than current (smaller) models.

For the study of privacy, our work indicates that current large language models memorize a significant fraction of their training datasets. Memorization scales log-linear with model size—by doubling the number of parameters in a model we can extract a significantly larger fraction of the dataset. Given that current state-of-the-art models contain more than $200\times$ as many parameters as the largest 6B parameter model we analyze, it is likely that these even larger models memorize many sequences

---

[3]We were unable to access the 175 billion parameter model; we run OPT models up to 66 billion parameters.

that are repeated just a handful of times. At the same time, we have shown that this memorization is often hard to discover, and for an attack to actually extract this data it will be necessary to develop qualitatively new attack strategies. Fortunately, it appears that (for the comparatively small models we study) training data inserted just once is rarely memorized, and so deduplicating training datasets (Lee et al., 2021) is likely a practical technique to mitigate the harms of memorization.

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

## A    IMPLEMENTATION DETAILS FOR DATASET CREATION

Intuitively speaking, it is straightforward to construct a dataset containing specifiable proportions of documents at various frequencies. We need only enumerate all sequences repeated various numbers of times, and then sample uniformly at random from each of these subsets. However in practice this is difficult to do, given the scale of these datasets: even asking the question "how many times is this sequence present in the training dataset" requires linear work for each query, and so repeating this thousands of times for an 800GB dataset would be infeasible.

To do this efficiently, we build on the work of Lee et al. (2021) and construct a suffix array over the training dataset. Such a data structure allows efficient queries to enumerate all sequences of length $k$ that are repeated between $N$ and $M$ times for any $N, M$. This can be accomplished by a linear scan of the suffix array. As notation, write $i$ as the pointer into the dataset at a certain position $j$ of the suffix array (i.e., $A[j] = i$), $i'$ as the index at position $j + N$ (so that $A[j + N] = i'$), and $i''$ as the index at position $j + M$ (so that $A[j + M] = i''$). Then, if $D[i : i + k] = D[i' : i' + k]$ but $D[i : i + k] \neq D[i'' : i'' + k]$, the sequence $D[k : i + k]$ is guaranteed to appear between $N$ and $M$ times in the dataset. As a result, we can scan linearly through the suffix array and enumerate all values of j $j$ to efficiently find all potential sequences repeated between N and M times. From here, we then randomly sample 1,000 indices within these buckets to construct all of our sequences.

## B    LONGER DOCUMENTS ARE NOT EASIER TO MEMORIZE THAN SHORTER DOCUMENTS

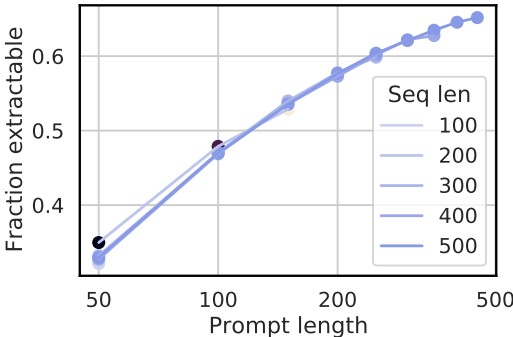

Figure 5: Longer sequences are not easier to extract. We compute the probability that an adversary can extract a sequence as a function of the number of tokens of context available, when varying the length of the sequences. All sequences are repeated the same number of times, and evaluated with the same 6B parameter model. Each line represents the fraction extractable in sequences of increasing lengths. Because all lines nearly perfectly overlap, longer sequences are not fundamentally "easier" to extract than shorter sequences.

Intuitively, one might think that longer sequences are more likely in the tail of the distribution, and if the model is trained to a low perplexity, then the tail of the distribution may be more likely to be memorized. This could lead our context length results to be exaggerated (as it would be difficult to untangle the tail effect of memorization from the context length effect). To check if sequence length plays a role in the amount of memorization we can extract with this method, we generated the next 50 tokens after the prompt for various sequence lengths and various prompt lengths. Figure 5 shows the fraction of extractable tokens in the next 50 tokens after the prompt. Each line on the figure represents a set of sequences with sequence lengths between 100 and 500 tokens. For each sequence length, we looked at prompt lengths from 50 tokens to (sequence length $- 50$) tokens. We do not see significant differences between the fraction of extractable tokens with varying prompt lengths across various sequence lengths.

| Prompt | Continuation (== 6B) | 2.7B | 1.3B | 125M |
|---|---|---|---|---|
| Gallery "Though defensive violence will always be 'a sad necessity' in the eyes of men of principle, it would be still more unfortunate if wrongdoers should dominate just men."- St. Augustine "A new idea is first | condemned as ridiculous, and then dismissed as trivial, until finally it becomes what everybody knows." - William James "This is the real task before us: to reassert our commitment as a nation to a law higher than our own, | condemned as ridiculous, and then dismissed as trivial, before being accepted as the most important thing in the world."- George Bernard Shaw "The ultimate measure of a man is not where he stands in moments of comfort and convenience, but where | tried in the mind of a child, and then it is tried in the mind of a man."- Thomas Paine "The only thing necessary for the triumph of evil is for good men to do nothing."- Edmund Burke " | and foremost a man's duty to his fellow man. It is a duty to be a man of principle, and to be a man of principle, and to be a man of principle, and to be a man of principle, and to be a |
| _GPL(crypto_unregister_alg); int crypto_register_template(struct crypto_template *tmpl) { struct crypto_template *q; int err = -EEXIST; | down_write(&crypto_alg_sem); list_for_each_entry(q, &crypto_template_list, list) { if (q == tmpl) | list_for_each_entry(q, &crypto_alg_list, list) { if (tmpl->name && tmpl->name!= q->alg.cra_name) | q = kzalloc(sizeof(*q), GFP_KERNEL); if (!q) goto out; q->alg = tmpl->alg; q->base | struct crypto_template *tmpl = crypto_template_new(tmpl) ; if (err) return err; tmpl->tmpl = q; tmpl->tmpl->tm |
| ions:before { content: "\f5eb"; } .fa-discord:before { content: "\f392"; } .fa-discourse:before { content: "\f393 | "; } .fa-divide:before { content: "\f529"; } .fa-dizzy:before { content: "\f567"; } .fa-dna:before | "; } .fa-digg:before { content: "\f391"; } .fa-dochub:before { content: "\f394"; } .fa-docker:before { | "; } .fa-digg:before { content: "\f96c"; } .fa-dollar-sign:before { content: "\f155"; } .fa-digniter | "; } .fa-discus:before { content: "\f394"; } .fa-drupal:before { content: "\f395"; } .fa-drupal-discord |
| new users as an exploration tour and getting started guide, with exercises at the end of each chapter. For more advanced trainees it can be a desktop reference, and a collection of the base knowledge needed to proceed with system and network administration. This | book contains many real life examples derived from the author's experience as a Linux system and network administrator, trainer and consultant. They hope these examples will help you to get a better understanding of the Linux system and that you feel encouraged to try out things on | book is designed to give the reader a firm understanding of the technologies needed to install and manage Linux systems, using the various available tools and techniques for the task. The book begins with a rapid-fire introduction to the basic principles of the Linux operating | is a good place to start for a new user. A: I would recommend the book "Linux Netw orking" by David S. It is a very good book for beginners. A: I would recommend | is a great way to get started with a new project. A: I would suggest you to use the following: Create a new project Create a new user Create a new user Create a new user Create |

Figure 6: Text examples that are memorized by the 6B model, but not by smaller models. Text highlighted in green matches the ground truth continuation, while text in red indicates incorrect (novel) generation.

## C  ALTERNATE EXPERIMENTAL SETTINGS

In this section, we study other strategies that we could have used to quantify memorization.

**Random dataset sampling.**  In Section 4.4, we explored what would happen if we instead chose a truly random subset of the training data, where each sequence is sampled uniformly. Specifically, we randomly sample 100,000 sequences from The Pile dataset of length 100, 200, 500, and 1,000; prompt the model with the first $N - 50$ tokens; and then test for memorization by verifying if the model can emit the remaining 50 tokens perfectly. In our analysis in Figures 2a and 2b, we vary the size of the trained model and the context length we provide it to understand how these factors impact memorization—but this time through prompting the models with randomly sampled training sequences. As expected, the absolute probability of memorization is much lower than in Figure 1 where we prompted models with training data from the sampled duplication-normalized subset.

We observe similar trends with model scale and context length as in our other results. Larger models memorize more training examples than smaller models—and much more than the baseline GPT-2 model that was not trained on The Pile. Similarly, providing more context to a model increases the likelihood we can discover memorization. In Figure 2b, we prompt models with: prompt length = sequence length − 50. We see that the longer prompts are easier to predict correctly than shorter prompts. The baseline GPT-2 model is nearly twice as accurate on sequences of length 1,000 (prompt length = 950) compared to sequences of length 100 (prompt length = 50).

**Alternate definition of extractability.**  Our main experiments report a sequence as "extractable" if the model's generated continuation is identical to the true suffix within that training example. This method is a loose lower bound on memorization. Consider two sequences $x_1$, $x_2$ both contained in the training dataset. Suppose these two sequences share the same prefix, and differ only in the final suffix; that is, $x_1 = [p||s_1]$ and $x_2 = [p||s_2]$. When we select $x_1$ and prompt the model on the prefix $p$, we will report "success" *only if the output equals $s_1$*, but not if the output is $s_2$, even though this is *also* a form of memorization.

We now consider how our results would change if we instead checked that the generation $[p||f(p)]$ from a prompt $p$ was contained *anywhere* in the training dataset. This gives a strictly larger measurement of memorization. By comparing these two methods (checking for memorization within the ground truth continuation, and within the entire dataset), we can understand how the choice of measurement affects the results in our experiments.

Searching within the entire dataset finds more memorized content than comparing with the ground truth (Figure 2c). For examples at 100 repetitions 32.6% of outputs are contained somewhere in the dataset but just 15.8% match the ground truth continuation. This difference becomes more pronounced as the number of repetitions increases. The maximum difference between these approaches is 28.4%, at 2,200 repetitions.

We refrain from using this approach for our main experiments, because this definition requires vastly larger computation resources; it requires querying whether hundreds of thousands of sequences are contained in an 800GB training dataset. Therefore, to promote reproducability, the remainder of this paper continues with testing the generated suffix against the single expected training suffix.

## D    TEXT MEMORIZED BY ONLY SOME MODELS

Table 1: The number of sequences memorized by one model, and not memorized by another.

| Model | Memorized | Not Memorized By | | | |
|---|---|---|---|---|---|
| | | 125M | 1.3B | 2.7B | 6B |
| 125M | 4,812 | - | 328 | 295 | 293 |
| 1.3B | 10,391 | 5,907 | - | 1,205 | 1,001 |
| 2.7B | 12,148 | 7,631 | 2,962 | - | 1,426 |
| 6B | 14,792 | 10,273 | 5,402 | 4,070 | - |

Table 1 shows the total number of sequences that are memorized by one model but not another. Larger models have more uniquely memorized sequences, although every model has some memorization not shared by any other model. (Even the 125M model memorizes a few sequences that the 6B model does not.)

## E    MEMORIZATION IN OPT MODELS

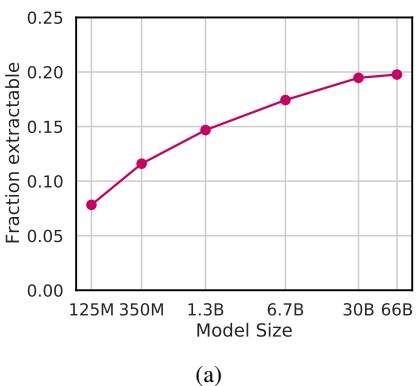

(a)

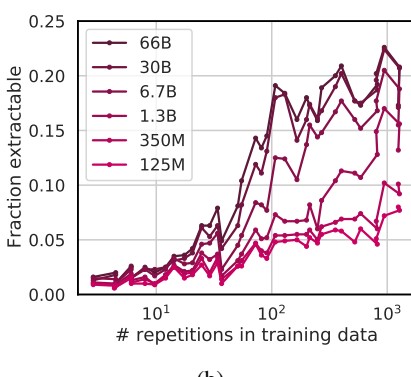

(b)

Figure 7: We prompt OPT models with data sampled from their training set. We use a prompt length of 100 here. **(a)** Fraction of sequences extracted as a function of model scale. **(b)** Fraction of sequences extracted as the number of repetitions of that sequence in the training set increases.

## F    EXAMPLES OF MEMORIZED TEXTS

We show examples of texts that are memorized by different models. We consider the case of 50-token prompts and 50-token generation. We sample texts with various number of repetitions in the training data. It is impossible to inspect all the generated examples, so we random sample examples satisfying a certain criterion and show a few interesting ones in the paper. Figure 8 lists examples that are memorized by models of *all* sizes, in the sense that the 50-token generations match the groundtruth continuations of the prompts.

| Prompt | Continuation (== 6B == 2.7B == 1.3B == 125M) |
|---|---|
| use this file except in compliance with the License. * You may obtain a copy of the License at * http://www.apache.org/licenses/LICENSE-2.0 * Unless required by applicable law or agreed | to in writing, software * distributed under the License is distributed on an "AS IS" BASIS, * WITHOUT WARRANTIES OR CONDITIONS OF ANY KIND, either express or implied. * See the License for the specific language |
| * * This program is free software; you can redistribute it and/or modify * it under the terms of the GNU General Public License version 2 and * only version 2 as published by the Free Software Foundation. * * | This program is distributed in the hope that it will be useful, * but WITHOUT ANY WARRANTY; without even the implied warranty of * MERCHANTABILITY or FITNESS FOR A PARTICULAR PURPOSE. See the * |
| Privacy & Cookies Policy Privacy Overview This website uses cookies to improve your experience while you navigate through the website. Out of these cookies, the cookies that are categorized as necessary are stored on your browser as they are essential for the working | of basic functionalities of the website. We also use third-party cookies that help us analyze and understand how you use this website. These cookies will be stored in your browser only with your consent. You also have the option to opt-out of |
| \end{document}$ in front of $\documentclass[12pt]{minimal} \usepackage{amsmath} | \usepackage{wasysym} \usepackage{amsfonts} |
| Len int for shift := uint(0); ; shift += 7 { if shift >= 64 { return ErrIntOverflowRaft | } if iNdEx >= l { return io.ErrUnexpectedEOF } b := dAtA[ |
| </object> <nil key="sourceID"/> <int key="maxID">18</int> </object> <object class="IBClassDescriber" key=" | IBDocument.Classes"> <object class="NSMutableArray" key="referencedPartialClassDescriptions"> <bool key="EncodedWithXMLCoder">YES</bool |

Figure 8: Text examples that are memorized by all the models: given 50-token prompts on the left, the next 50 tokens generated by all the models match the groundtruth continuation.

Figure 9 lists examples that are memorized by the 6B model but not by smaller ones. Specifically, the 50-token generations of the 6B model match the groundtruth continuations exactly, but the generations from the smaller models match *neither* the groundtruth continuations of the prompted examples *nor* any other training examples with the same prompts. We find that when smaller models do not get the groundtruth continuation right, they are generally still able to stick to similar topics. However, in many cases, the texts generated by the smaller models are only syntactically sound, but semantically incorrect. Figure 10 and Figure 11 show more examples.

In Figure 12 we show examples that are only memorized by the smallest model, using similar criterion as when we filter examples that are only memorized by the largest model. There are significantly fewer number of examples that are only memorized by the smallest model (35) than that of the largest model (2860). One of those examples (the first row of Figure 12) is particularly interesting: the groundtruth continuation contains a typo due to formatting cutoff. While the smallest model memorized the typo, larger models try to fix the typo.

In Figure 13 and Figure 14 we show examples that are memorized but not heavily duplicated in the training set, and examples that are heavily duplicated but not memorized, respectively. Finally, we show examples that are memorized by GPT2-XL in Figure 15.

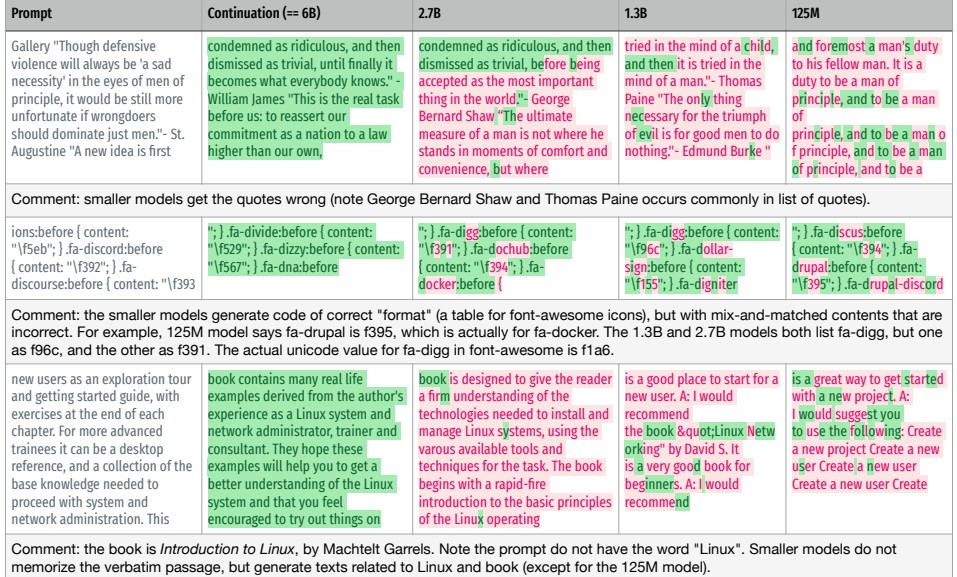

Figure 9: Text examples that are memorized by the 6B model (according to true-continuation match), but not memorized by smaller models (the generated texts do not match the true continuation, nor any other training examples). The first column shows the prompt. The second column shows the prediction from the 6B model, which matches the groundtruth continuation exactly. The remaining columns shows predictions from smaller models.

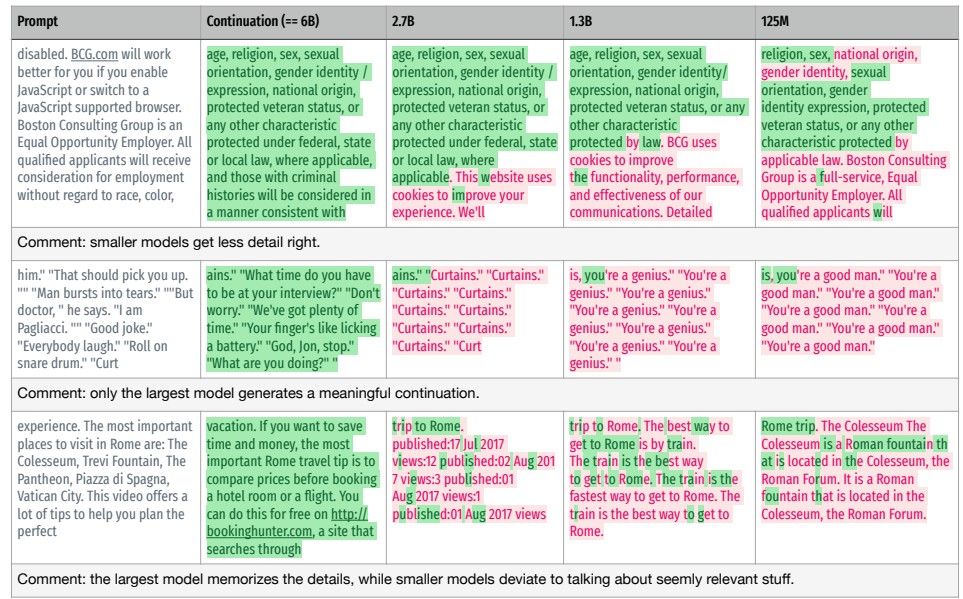

Figure 10: Continuation of Figure 9.

| Prompt | Continuation (== 6B) | 2.7B | 1.3B | 125M |
|---|---|---|---|---|
| _GPL(crypto_unregister_alg); int crypto_register_template(struct crypto_template *tmpl) { struct crypto_template *q; int err = -EEXIST; | down_write(&crypto_alg_sem); list_for_each_entry(q, &crypto_template_list, list) { if (q == tmpl) | list_for_each_entry(q, &crypto_alg_list, list) { if (tmpl->name && tmpl->name! = q->alg.cra_name) | q = kzalloc(sizeof(*q), GFP_KERNEL); if (!q) goto out; q->alg = tmpl->alg; q->base | struct crypto_template *tmpl = crypto_template_new(tmpl); if (err) return err; tmpl->tmpl = q; tmpl->tmpl->tm |
| Comment: smaller models do not generate the exact code, but the variables refer to names in the prompt. | | | | |
| , such as Google, Bing and Yahoo!, use crawlers to find pages for their algorithmic search results. Pages that are linked from other search engine indexed pages do not need to be submitted because they are found automatically. The Yahoo! Directory and D | MOZ, two major directories which closed in 2014 and 2017 respectively, both required manual submission and human editorial review.[39] Google offers Google Search Console, for which an XML Sitemap feed can be created and submitted for free to ensure that all | MOZ are two examples of directories that use this technique. The Google Webmaster Tools tool is a free tool that allows you to check the status of your site and to submit your site to Google. The Google Webmaster Tools tool | MOZ are examples of such directories. The search engines are also used to find pages that are not indexed by the search engines. These pages are referred to as "orphaned" pages. The search engines are also used to find pages that | igg search engine are also used to find pages for their algorithmic search results. The Google search engine is a search engine that uses Google's search engine to find pages for its algorithmic search results. The Google search engine uses the Google Search |
| Comment: smaller models talk about relevant topics but are not as coherent as 6B model (which memorizes the training data verbatim). | | | | |
| be 21 years of age to purchase this item. Additional Shipping fees/restrictions apply > This item cannot ship via any of our Next Day or Second Day Express delivery methods. Shipping restrictions apply to this product under some | jurisdictions. Known age, state and local shipping restrictions are applied at checkout and may result in changes to an order. PLEASE NOTE: We do not accept returns of Ammunition for safety reasons. NEVER use any Ammunition unless it is specifically designed for your firearm. | jurisdictions. Known age, state and local shipping restrictions are applied at checkout and may result in changes to an order. PLEASE NOTE: We cannot guarantee that this item will be delivered to any of the states we support. This item cannot ship to | jurisdictions. Please check with your local dealer or retailer for specific shipping restrictions on this product. This item cannot ship to Canada. This item cannot ship internationally. This item cannot be gift boxed or shipped. | jurisdictions. This item cannot ship via any of our Next Day or Second Day Express delivery methods. Shipping restrictions apply to this product under some jurisdictions. This item cannot ship via any of our Next Day or Second Day Express delivery |
| Comment: smaller models talk vaguely about shipping restrictions (125M model simply repeat the prompts). | | | | |

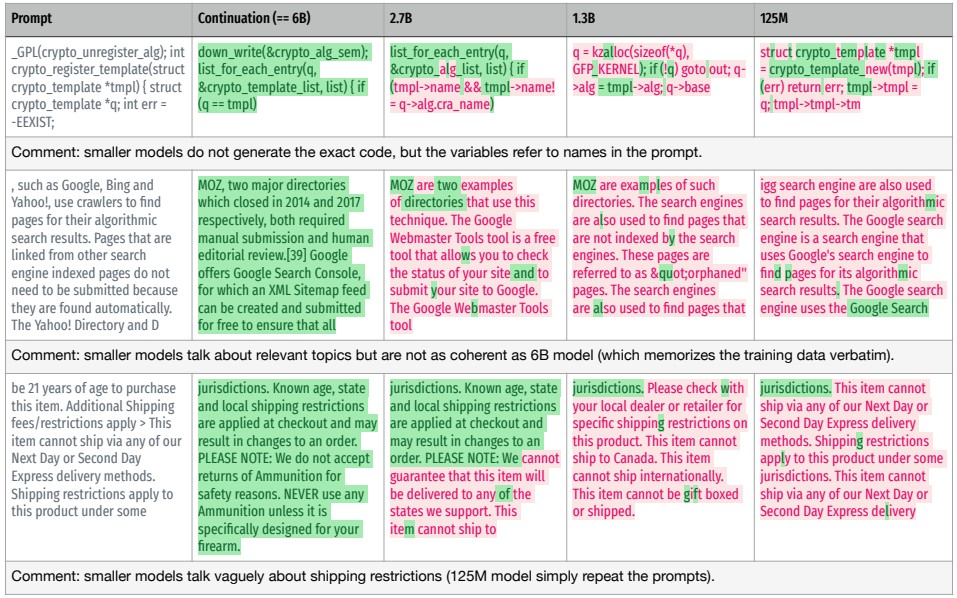

Figure 11: Continuation of Figure 9.

| Prompt | 6B | 2.7B | 1.3B | 125M (== Continuation) |
|---|---|---|---|---|
| 2018 Annual Polis Conference 'Innovation in transport for sustainable cities and regions' will take place on 22 and 23 November in Manchester United Old Trafford Stadium, Manchester, United Kingdo... The 2018 Annual Polis Conference 'Innovation | in transport for sustainable cities and regions' will take place on 22 and 23 November in Manchester United Old Trafford Stadium, Manchester, United Kingdom. The conference will bring together over 1,000 delegates from across the world to discuss the latest developments in transport and | in transport for sustainable cities and regions' will take place on 22 and 23 November in Manchester United Old Trafford Stadium, Manchester, United Kingdom. The conference will bring together transport and urban planning experts, policy makers, academics, practitioners and students to | in transport for sustainable cities and regions' will take place on 22 and 23 November in Manchester United Old Trafford Stadium, Manchester, United Kingdon... The 2018 Annual Polis Conference 'Innovation in transport for sustainable cities and regions' will | in transport for sustainable cities and regions' will take place on 22 and 23 November in Manchester United Old Trafford Stadium, Manchester, United Kingdo... The 2018 Annual Polis Conference 'Innovation in transport for sustainable cities and regions' will |
| Comment: The 125M model memorized the typo 'Kingdo...'; the 1.3B model fixed the typo but to a (semantically) wrong word. The 2.7B and 6B models fix this word correctly. | | | | |
| m_Scale: {x: 1, y: 1} m_Offset: {x: 0, y: 0} - _MainTex: | m_Texture: {fileID: 2800000, guid: ee9e9e9 e9e9e9e9e9e9e9e9e9e9e9e9e9e9 | m_Texture: {fileID: 2800000, guid: ae8b9e8b 9a8e4b4a8b9b9b9b9b9b9b, type | m_Texture: {fileID: 2800000, guid: bf9a8d9c9a 9a4a4a9b8d9d8d6e9d9d9, | m_Texture: {fileID: 0} m_Scale: {x: 1, y: 1} m_Offset: {x |
| Comment: the smallest model memorized the code, while the larger models try to generate some GUIDs. | | | | |
| NOT LIMITED TO, PROCUREMENT OF SUBSTITUTE GOODS OR SERVICES; * LOSS OF USE, DATA, OR PROFITS; OR BUSINESS INTERRUPTION) * HOWEVER CA | USED AND ON ANY THEORY OF LIABILITY, WHETHER IN CONTRACT, * STRICT LIABILITY, OR TORT (INCLUDING NEGLIGENCE OR OTHERWISE) * ARIS | USED AND ON ANY THEORY OF LIABILITY, WHETHER IN CONTRACT, * STRICT LIABILITY, OR TORT (INCLUDING NEGLIGENCE OR OTHERWISE) * ARIS | USED AND ON ANY THEORY OF LIABILITY, WHETHER IN CONTRACT, * STRICT LIABILITY, OR TORT (INCLUDING NEGLIGENCE OR OTHERWISE) * ARIS | USED AND ON ANY THEORY OF LIABILITY, WHETHER IN * CONTRACT, STRICT LIABILITY, OR TORT (INCLUDING NEGLIGENCE * OR OTHERWISE) ARIS |
| Comment: the larger models generate a slight variation of the license text. Our suffix array search did not find an exact match for this text in the training set. | | | | |

Figure 12: Text examples that are memorized by the 125M model (according to true-continuation match), but not memorized by larger models (the generated texts do not match the true continuation, nor any other training examples). The first column shows the prompt. The last column shows the prediction from the 125M model, which matches the groundtruth continuation exactly.

| Frequency | Prompt | Continuation ( == 6B) |
|---|---|---|
| 2 | L_LONG_LONG */ __STL_TEMPLATE_NULL struct __type_traits<float> { typedef __true_type has_trivial_default_ | constructor; typedef __true_type has_trivial_copy_constructor; typedef __true_type has_trivial_assignment_operator; |
| 2 | "groupby4_map", "groupby4_map_skew", "groupby4_noskew", "groupby5", | "groupby5_map", "groupby5_map_skew", "groupby5_noskew", "groupby6", |
| 2 | pair<K12>(_12), fusion::make_pair<K13>(_13), fusion::make_pair<K14>(_14), fusion::make_pair<K15>(_15)); | } namespace result_of { template < typename K0, typename K1, typename K |
| 2 | _GLSL_400)) list += QLatin1String("dmat2"); if (variant & (Lexer::Variant_GLSL_400)) list += QLatin | 1String("dmat3"); if (variant & (Lexer::Variant_GLSL_400)) list += QLatin1String("dmat4"); if (variant & ( |
| 3 | disabled. BCG.com will work better for you if you enable JavaScript or switch to a JavaScript supported browser. Boston Consulting Group is an Equal Opportunity Employer. All qualified applicants will receive consideration for employment without regard to race, color, | age, religion, sex, sexual orientation, gender identity / expression, national origin, protected veteran status, or any other characteristic protected under federal, state or local law, where applicable, and those with criminal histories will be considered in a manner consistent with |

Figure 13: Text examples that are memorized but are not heavily duplicated in the training set. Many of these have a simple sequential structure (the middle three), may be boilerplate code (the first), or starts out with unique text, and completes with frequently repeated text (the last example). Overall, these are easily completed sequences.

| Frequency | Prompt | Continuation | 6B |
|---|---|---|---|
| 5526 | contributors. All rights reserved. // // Licensed under the Apache License, Version 2.0 (the "License"); // you may not use this file except in compliance with the License. // You may obtain a copy of the | License at // http://www.apache.org/licenses/LICENSE-2.0 // // Unless required by applicable law or agreed to in writing, software // distributed under the License is distributed on an "AS | License at // // http://www.apache.org/licenses/LICENSE-2.0 // // Unless required by applicable law or agreed to in writing, software // distributed under the License |
| 5652 | ators are Standing By Our Dumpster Specialists are waiting by the phone to help you get started. However, there is some important information we need before we can give you a price quote and delivery time. Some things we need to know | : What size dumpster you will need What you plan on putting in your dumpster When you need us to drop your roll off container off How long you plan on keeping your roll-off City & | include: What size dumpster do you need? What type of waste do you have? How much waste do you have? What is the delivery time? What is the pickup time? What |
| 4323 | a copy of the License at # # http://www.apache.org/licenses/LICENSE-2.0 # # Unless required by applicable law | or agreed to in writing, software # distributed under the License is distributed on an "AS IS" BASIS, WITHOUT # WARRANTIES OR CONDITIONS OF ANY KIND, either express or implied. See | or agreed to in writing, software # distributed under the License is distributed on an "AS IS" BASIS, # WITHOUT WARRANTIES OR CONDITIONS OF ANY KIND, either express or implied. |
| 3556 | date_default_timezone_set() function. In case you used any of those methods and you are still getting this warning, you most likely misspelled the timezone identifier. We selected the timezone 'UTC' for now, but please | set date.timezone to select your timezone. in /home/erlypro/public_html/natural/datas/persobanner_center.php on line 17 Deprecated: Function eregi_replace | set date.timezone to select your timezone. in /home/notes/public_html/wp-includes/functions.php on line 3812 Warning: date() [function.date]: It is not safe to rely |
| 3920 | \|   [手把手翻墙教程](https://github.com/gfw-breaker/guides/wiki)  & | nbsp;\|   [禁闻聚合安卓版](https://github.com/gfw-breaker/bn-android)  &n | nbsp; [网门安卓版](https://github.com/gfw-breaker/webpack-interpreter)    [神州正道 |
| 3675 | * / _\|\|\|\|)\|\|* | \|(_\|\|_\|\|_◁\|___* | \|(_\|\|_\|\|_\|\|* |
| 3030 | 1775, "date": "1775\u20131851", "displayOrder": 1, "fc": "Joseph Mallord William | Turner", "gender": "Male", "id": 558, "mda": "Turner, Joseph Mallord William", | Turner", "gender": "Male", "id": 604, "mda": "Turner, Joseph Mallord William", |

Figure 14: Text examples that are heavily replicated in the training set but not memorized. We find many examples which have slight differences with no semantic (English) meaning. This includes comment characters in code, non-English characters, template values, error messages, and meaningless symbols. We also surprisingly find a large number of slightly different but heavily repeated documents about dumpsters.

| Prompt | Continuation (== GPT2-XL) |
|---|---|
| 165.00 $175.00 $180.00 $185.00 $190.00 $195.00 $200.00 $210.00 $220.00 $230.00 $240.00 $250.00 $260. | 00 $270.00 $280.00 $290.00 $300.00 $310.00 $320.00 $330.00 $340.00 $350.00 $360.00 $370.00 $380.00 $ |
| ISRC3DEC2, ARIZONA_ISRC3DEC2MIX_INPUT_1_SOURCE); ARIZONA_MUX_ENUMS(ISRC3DEC3, ARIZONA_ | ISRC3DEC3MIX_INPUT_1_SOURCE); ARIZONA_MUX_ENUMS(ISRC3DEC4, ARIZONA_ISRC3DEC4MIX_INPUT_ |
| the councils of government, we must guard against the acquisition of unwarranted influence, whether sought or unsought, by the militaryindustrial complex. The potential for the disastrous rise of misplaced power exists and will persist. We must never let the weight | of this combination endanger our liberties or democratic processes. We should take nothing for granted. Only an alert and knowledgeable citizenry can compel the proper meshing of the huge industrial and military machinery of defense with our peaceful methods and goals, so that security and |
| full spreadsheet NEW! Buy our book • Facts are Sacred: the power of data (on Kindle) More open data Data journalism and data visualisations from the Guardian World government data • Search the | world's government data with our gateway Development and aid data • Search the world's global development data with our gateway Can you do something with this data? • Flickr Please post your visualisations and mash-ups on |
| Original press release Get ahead of the crowd by signing up for 420 Investor, the largest & most comprehensive premium subscription service for cannabis traders and investors since 2013. Published by NCV Newswire The NCV Newswire | by New Cannabis Ventures aims to curate high quality content and information about leading cannabis companies to help our readers filter out the noise and to stay on top of the most important cannabis business news. The NCV Newswire is hand-curated by |
| of sole responsibility of the author(s). The Centre for Research on Globalization will not be responsible for any inaccurate or incorrect statement in this article. The Centre of Research on Globalization grants permission to cross-post Global Research articles on community internet sites | as long the source and copyright are acknowledged together with a hyperlink to the original Global Research article. For publication of Global Research articles in print or other forms including commercial internet sites, contact: [email protected] www.globalresearch.ca |

Figure 15: Text examples that are from The Pile and memorized by GPT2-XL. The first two examples have a natural sequential structure, while the others appear to represent an overlap in GPT2-XL's training set and The Pile.

