# OpenReview forum: "Quantifying Memorization Across Neural Language Models"
_ICLR.cc/2023/Conference — ICLR 2023 notable top 25%_

### Official Review · Reviewer_6k5t · 2022-10-23

**Confidence:** 4
**Correctness:** 3
**Technical Novelty And Significance:** 2
**Empirical Novelty And Significance:** 2
**Recommendation:** 6

**Clarity, Quality, Novelty And Reproducibility:**

The conclusion of the paper is that larger models, more duplication and longer prompts all increase the likelihood of reproducing the training example. All of these properties have been investigated before by previous work, and several of these works are also cited in this paper. The experiments are somewhat more thorough than previous work (although not without issues), but neither the hypotheses nor the findings are novel. Therefore the novelty of the paper seems to be limited to scaling up previous diagnostic experiments.

One of the contributions highlighted in the introduction is the finding that GPT-J memorizes at least 1% of its dataset, compared to a previous lower measurement for GPT-2.
However, these are two different models, and more importantly they have been trained on different datasets. The experiments show that the number of duplicates present in the training set leads to more examples being memorized. So it seems that the simplest explanation to GPT-J memorizing more than GPT-2 would be that the training data for GPT-J contained more duplicates.

The main set of results are presented on a dataset that controls both the distribution of duplication and the distribution of length. This would be suitable for presenting results at different duplication and length levels, but the results with different model size or prompt length become less meaningful due to the artificial nature of the distributions. Furthermore, because the data is controlled for two distributions at the same time, it is unclear how these two distributions interact and what kind of artefacts that may create. Luckily, results on a random distribution are also reported as auxiliary experiments.

GPT-2 is reported as a baseline on the same dataset of examples from the training set of GPT-J. However, it is a different model trained on a different dataset. We also don't know what percentage of those particular examples were present in the GPT-2 training set and how many times they were duplicated there, so it's unclear what the baseline is meant to show.

Experiments with other models (e.g. T5) find lower rates of memorization depending on the duplication counts. But these experiments are performed on the same data that was constructed specifically from the duplicated examples of GPT-J? It seems that result is explained simply by those examples not being duplicated in the T5 training data.

When experimenting with different prompt lengths, is seems different sets of examples are chosen so that they would match the correct prompt length. That would mean the results are on different texts, and the prompt length is not the only thing that is varied. There may be other properties of the texts that are correlated with their length, particularly if that dataset also favours highly duplicated examples.

The practical applicability of these experiments needs some more discussion. First, it's unclear what would be the realistic scenario where an attacker would know 100+ exact tokens from the training set but not the next token. The paper also discusses that the results are skewed by correctly generating long sequences of whitespace, which doesn't seem like a practical case of memorization.

It is said that most models use random sampling to generate, but only greedy sampling is investigated in this paper. It seems results with random sampling would then have more practical usefulness. The choice of restricting to greedy sampling could at least be motivated more.

**Strength And Weaknesses:**

Strengths:

* Investigation of the foundational language models is an important topic.
* The experiments are systematic and more thorough than previous work.

Weaknesses:

* The same properties of pre-trained language models have all been investigated and demonstrated by previous work. There don't seem to be any novel hypotheses or findings.
* Some of the experiments have questionable setups (more details below).

**Summary Of The Paper:**

The paper investigates the reproduction of training examples from pre-trained language models.
The amount of memorization is measured across 1) different model sizes, 2) the number of times an example is duplicated in the training data, 3) the length of prompt given as input.
It is concluded that larger models, more duplication and longer prompt all increase the likelihood of reproducing the training example.


**Summary Of The Review:**

The investigated area is important and the experiments are more thorough than in previous work.
However, the novelty of the experiments and the findings is limited and it is not clear that all the experiments show what they are claiming to show.

---

> ### Author Response · Authors · 2022-11-18
> **Author Response**
>
> > All of these properties have been investigated before by previous work
>
> We agree that prior work has anecdotal evidence that model scale and prompt length contribute to memorization, but to the best of our knowledge no serious study has been done of these factors. If the reviewer is aware of any study of this nature please let us know.
>
> > but neither the hypotheses nor the findings are novel
>
> We respectfully disagree. A finding does not need to be “surprising” to be novel. Anecdotally the research community believed many of the findings we show, but we actually do the work to demonstrate they are true across settings. This is important and novel even if unsurprising.
>
> > So it seems that the simplest explanation to GPT-J memorizing more than GPT-2 would be that the training data for GPT-J contained more duplicates
>
> If the reviewer honestly believes that the conclusion of our paper is that GPT-J memorizes 1,000,000x as much data as GPT-2 then we believe that this is a massively novel and significant finding. We believe the much more likely observation is that prior work *undercounted* memorization by 1,000,000x — especially because this prior work explicitly said so: “Note that this is likely an extremely loose lower bound. We only manually inspected 1,800 potential candidate memorized samples; if we had started with more candidates we would likely have identified significantly more memorized content”.
>
> > There may be other properties of the texts that are correlated with their length, particularly if that dataset also favours highly duplicated examples.
>
> We show in the appendix that length does not factor into difficulty of examples.

---

> > ### Comment · Reviewer_6k5t · 2022-11-23
> > **Reviewer response**
> >
> > You repeatedly refer to previous work as "anecdotal", in a seeming attempt to belittle it. I would say the experiments by Carlini et al are quite thorough, they just took a different approach.
> >
> > I will slightly raise my score for this paper as it has some experiments that may provide useful information.
> > However, the comparisons to previous work and the comparisons to other models are still highly problematic.
> > Carlini et al investigated memorisation by either prompting on a) only the sentence start token, or b) randomly crawled internet text.
> > In contrast, your experiments prompt with 50 tokens directly from the training data. Therefore the difficulty of reproduction is much lower in your setting, so the higher success rate is expected. Taken to the limit, one could prompt based on the full sequence and only generate one word for completion, and the success rate would even be much higher.
> >
> > Any assertions about memorisation in other models besides GPT-J in this paper (e.g. using them as baselines) are not valid because they do not consider the training data of those models, therefore there is no way to know if a particular example was memorised or not.

---

> > > ### Author Response · Authors · 2022-11-23
> > > **Clarification our comparison to prior work**
> > >
> > > We do not intend to belittle prior work---we quite like the work of Carlini et al. and our paper is a direct extension of theirs.
> > >
> > > Let's recall what experiment Carlini et al. (2021) performs. They find a single training document that was contained in the GPT-2 training dataset, that repeated 13 URLs either 8, 17, 33, 35, 51, 56, 64, 72, 76, 113, or 359 times. They find GPT-2 XL memorized more of these than GPT-2 Medium which memorizes more than GPT-2 Small. So they've done 13 x 3 = 39 total pairs of model evaluations to understand memorization, and from this they draw two predictions about scale and duplications. These predictions appear true! But it actually does need to be shown empirically---and that's what we're doing here.
> > >
> > > In order to do this, we run ~1,000,000 sequences through models that are even 6x larger across several categories of neural networks. This is what gives us the confidence to say that we actually believe these scaling hypotheses.
> > >
> > > Finally, the other major difference between our two papers is that Carlini et al. (2021) is computing a *lower* bound of memorization, and we are computing an *upper* bound of memorization assuming full adversarial knowledge to "quantify the memorization" even if it's not possible that an attack could recover this memorization.

---

> > > > ### Comment · Reviewer_6k5t · 2022-11-23
> > > > **Reviewer response**
> > > >
> > > > Absolutely nowhere in your paper or in your previous comments have you previously indicated that your estimates are an **upper** bound. Quite the contrary, the paper seems to claim to present a new lower bound: "memorizes **at least** 1% of its training dataset".
> > > >
> > > > Please update your paper accordingly.

---

> > > > > ### Author Response · Authors · 2022-11-23
> > > > > **Bounds; upper and lower**
> > > > >
> > > > > Sorry let us clarify: there are two kinds of bounds here. Both of the statements are true at the same time.
> > > > >
> > > > > One bound is "upper/lower bound for what an attack can achieve". This is what we meant in the comment you are replying to.
> > > > > - A lower bound (Carlini et al. 2021) can be demonstrated by a practical attack. This paper sets a lower bound of ~600 memorized examples in the 40GB dataset.
> > > > > - An upper bound (our paper) can be demonstrated by showing what the model memorizes by sampling with long context, and then we know that the amount you could extract with the Carlini et al. (2021) attack must be lower because it has access to strictly less prompt information.
> > > > >
> > > > > Another bound is "upper/lower bounds of total amount of memorization". This is what we meant in the introduction to the paper.
> > > > > - A lower bound can be demonstrated by showing that there exists a strategy to use lots of context, with arg-max decoding, to recover training data. This is what we have done---show that the model does memorize at least 1% of its training dataset.
> > > > > - An upper bound could be demonstrated by some other approach, but we don't try to do this. We try to show a lower bound on what the model definitely memorizes.

---

> > > > > > ### Comment · Reviewer_6k5t · 2022-11-24
> > > > > > **Reviewer comment**
> > > > > >
> > > > > > I agree with the part about "upper/lower bound for what an attack can achieve". And this difference from Carlini et al should be clarified in the paper.
> > > > > >
> > > > > > I am not sure I can agree with the lower bound on memorization.
> > > > > > If I give 100 tokens of context and show 2% success at recovering the rest, is this now an "improved" lower bound on memorization? Or if I give n-1 tokens as input, correctly predict the last token 80% of the time, is this a lower bound as well?

---

> > > > > > > ### Author Response · Authors · 2022-11-24
> > > > > > > **Yes that is what a lower bound means**
> > > > > > >
> > > > > > > When we say something is a "lower bound" we mean it bounds the true number from below. For example, if an algorithm takes exactly 3 * n^2 operations to complete, then a first paper might slow a lower bound of 3 * n operations, another paper might improve this lower bound to max(n^2, 3 * n), and a final paper might give a tight bound of 3 * n^2.
> > > > > > >
> > > > > > > Similarly, here, we have shown a lower bound of 1% of the dataset that we can extract. But future work might be able to do better, and if some future paper shows that with some improved algorithm they can recover 2% of the dataset they will have improved on our lower bound.
> > > > > > >
> > > > > > > Now to talk precisely about the suggestion you are giving here: one has to be careful when computing the 1% number: it is invalid to count the prompt length in this measurement. So for example, if we show that we can predict the final 1 token of every document in the dataset given the n-1 prior tokens, the total fraction of data we have extracted from this model will be just (#documents)/(#tokens), which for a dataset like the Pile is <<1%. So that specific method wouldn't actually give a tighter bound.
> > > > > > >
> > > > > > > The tightest bound we could build using our techniques would be to compute the fraction of the dataset you could extract given exactly the prior n-1 tokens for every value of n. But this would be quadratically hard to estimate. So instead we compute this number by rounding down to the multiple next lowest multiple of 50. So for example, we consider the first 50 tokens of every document as impossible to extract, and measure the difficulty of extracting tokens 50-99 assuming the adversary only has access to tokens 0-49, etc. So what we're measuring already subsumes the suggestion you've given of prompting with 100 tokens.
> > > > > > >
> > > > > > > So, if future work was able to show a tighter lower bound by measuring with a more refined analysis technique we would view that as progress. We're not claiming that our bounds are tight in any way. (For example, we show that using beam search might give an even better lower bound.) But we believe our bound is useful.

---

> > > > > > > > ### Comment · Reviewer_6k5t · 2022-11-24
> > > > > > > > **Reviewer response**
> > > > > > > >
> > > > > > > > I am well aware of what the definition of a lower bound is, thank you.
> > > > > > > >
> > > > > > > > "So instead we compute this number by rounding down to the multiple next lowest multiple of 50. So for example, we consider the first 50 tokens of every document as impossible to extract, and measure the difficulty of extracting tokens 50-99 assuming the adversary only has access to tokens 0-49, etc. So what we're measuring already subsumes the suggestion you've given of prompting with 100 tokens."
> > > > > > > >
> > > > > > > > This is some really weird mental gymnastics.
> > > > > > > >
> > > > > > > > Surely you agree that giving more context as input makes it more likely for the language model to predict the correct next word? Using 50 tokens as context, you got a higher memorization percentage than Carlini et al, who used 0 tokens of context. And if I ran a new experiment with 100 tokens of context, I would most likely get a higher percentage than your 1%.
> > > > > > > >
> > > > > > > > Similarly, having to generate only the 50 tokens of a sequence (as you chose to do) is easier than Carlini et al, who tried to generate full sequences of unknown length. And if I ran an experiment that measures the generation of only 1 token, then that will have a much higher success rate than 1%.
> > > > > > > >
> > > > > > > > If I now run your experiments using 100 tokens of context and 1 token to generate, it will likely get around 50% accuracy. Surely you can't say that this means the model has memorized 50% of the training data (or that it's a lower bound for memorization).
> > > > > > > > So why exactly is your chosen 50 tokens of context and 50 tokens to generate a useful lower bound for training data memorization?

---

> > > > > > > > > ### Author Response · Authors · 2022-11-24
> > > > > > > > > **Clarifying definitions**
> > > > > > > > >
> > > > > > > > > There are two numbers here that matter. The first is the amount of context, which we vary from 50 tokens to 450 tokens. The second is how many tokens have to be extracted for us to count this as "memorization", and following Lee et al. (2022) we set this to 50 tokens. The reason we pick 50 tokens and not, say, 1 token is that we agree with Lee et al. and do not believe generating 1 token at a time counts as "memorization" because it could just as well be generalization.
> > > > > > > > >
> > > > > > > > > Given this, the number we report as a lower bound is the following: "given any strategy, what fraction of the training data can be perfectly recovered as part of a 50-token substring?"
> > > > > > > > >
> > > > > > > > > We lower bound this by basically asking the question "Given all of the context up to this token, can we complete the 50 following tokens verbatim?" Except instead of considering all possible contexts, we pick the longest context from among [50, 100, 150, ..., 450] so that we don't have to do quadratic work. We believe stopping at 450 is valid because it looks like (from our scaling curves) there are diminishing returns after this point.
> > > > > > > > >
> > > > > > > > > Certainly we agree that our task is easier than giving 0 tokens of context. The comment we are trying to make above is that we're not setting a context of just 50 tokens as some arbitrary number (as opposed to 100 as you say, or anything else) which would be very hard to justify; rather, we are trying to give the most context possible in order to compute this metric.
> > > > > > > > >
> > > > > > > > > We of course agree this is a hyperparameter of 50-tokens-to-count-as-memorization could vary. We've set it at a very conservative value where we hope most researchers would agree that 50 tokens of text being a verbatim match means it was memorized. But if some future paper picked a smaller value for this constant, then the numbers would be incomparable because the definitions would not match. And if some paper wanted to set it to 1, observed that 50% of the time we could predict this correctly, and wanted to argue this meant 50% of the data was extractable they could try and make this argument. But in our paper we don't do this.

---

> > > > > > > > > > ### Comment · Reviewer_6k5t · 2022-11-24
> > > > > > > > > > **Reviewer comment**
> > > > > > > > > >
> > > > > > > > > > I do agree with most of this. My point is that these are all important caveats and specifications of your work which should be made clear in the paper.
> > > > > > > > > >
> > > > > > > > > > The paper currently presents the 1% finding as a tighter lower bound compared to previous work, such as Carlini et al.
> > > > > > > > > > When in fact you have redefined the task and made it easier. Just as another paper using different amounts of context and generation target amounts would not be comparable to your results, your results are not comparable to the previous work.
> > > > > > > > > >
> > > > > > > > > > I do think your experiments as-is have value. I just disagree with how they are framed in the context of previous work, both in terms of comparing to previous results and making assertions about memorisation of other models besides GPT-J.

---

### Official Review · Reviewer_CQAU · 2022-10-24

**Confidence:** 4
**Correctness:** 3
**Technical Novelty And Significance:** 1
**Empirical Novelty And Significance:** 2
**Recommendation:** 5

**Clarity, Quality, Novelty And Reproducibility:**

This paper provides insights to pre-training language models: model can memorize even when using deduplicated training data.
As pointed in the weakness section, it's not clear how a substring is defined in this paper and how this relates to deduplication method. Thus, I find this paper lacking guidance or insights on how we can reduce repetition, which makes this paper not a huge significant contribution compared to previous papers that also reveal similar observations.

A missing reference: Memorization Without Overfitting: Analyzing the
Training Dynamics of Large Language Models, as the other important dimension when measuring memorization is the order of training data presented to the language models.

**Strength And Weaknesses:**

**Strength**
1. Paper is very well written and easy to understand.
2. The experiments conducted in this paper are quite extensive and results are convincing.
3. I appreciate that authors perform experiments on recently released models, e.g. OPT, although its training data is not fully transparent to the public.

**Weakness**
1. Duplication is a very important factor in the paper when measuring memorization, but I didn't find how a sequence of length l is obtained /extracted from the training data. Did you consider all the substrings of length l in the training data? Did the segmentation you used to obtain substrings match that of those deduplication methods used for pre-training (section 5.2)? Are there overlap between sequences of length 100 and sequences of length 200?
2. As discussed in the paper, the biased sampling method normalized by length and duplication times is more biased towards duplicated sequences, although there are experiments on uniformly sampled subsets, I think doing experiments on a subsets without any duplications is necessary. Again, duplication is dependent on how the substring is defined, which is worth discussing.
3. The definition of memorization is pretty strict in this paper, but in reality a paraphrased continuation which might not exactly exist in training corpus can still leak data. I think studying the generation in a relaxed setting is more interesting, e.g. consider using paraphrased prefixes or perturb the prefixes, as considering the paraphrased continuation can be hard to quantify the extractability.
4. Although authors provide many qualitative memorization examples, it's still hard for me to have a clear idea what types of sequences are more easily extracted, I think a dedicated section

**Summary Of The Paper:**

This paper extensively studies the memorization of training data in neural language models. First, it defines memorization as "given a prefix, the model can generation a continuation that exists in the training data". Then, it focuses on three aspects that affect a model's memorization rate: duplication times, length of prefix and model scales. It draws three main conclusions from a series experiments: (1) bigger models memorize more; (2) data that has more duplicates are memorized more; (3) longer prefixes can more easily extract memorized text. Furthermore, authors extend their experiments to other types of language models and models that are trained on deduplicated training data.

**Summary Of The Review:**

The empirical results presents in this paper provide insights to pre-training language models: model can memorize even when using deduplicated training data. But it lacks guidance or insights on how to reduce memorization and the explicit complications brought by memorization. Thus I feel that this paper has marginal contributions.

---

> ### Author Response · Authors · 2022-11-18
> **Author Response**
>
> We address each of your questions below:
>
> > Did you consider all the substrings of length l in the training data?
>
> We randomly sampled from among all strings of length l to make the problem computationally tractable. Our results have margins of error of near-zero given that we sample ~100,000 sequences.
>
> > Did the segmentation you used to obtain substrings match that of those deduplication methods used for pre-training (section 5.2)?
>
> During train set deduplication we used an approximate deduplication method, and to find repeated sequences we used an exact duplicate matching algorithm.
>
> > Are there overlap between sequences of length 100 and sequences of length 200?
>
> Yes.
>
> > I think doing experiments on a subsets without any duplications is necessary
>
> We agree and that’s why we also ran experiments on a uniform sampling of the data (Figure 2). Does the reviewer have suggestions for other experiments to run?
>
> > The definition of memorization is pretty strict in this paper
>
> We agree, however here we follow in line with prior work that studies exact duplication (Carlini et al. 2020). We hope that future work will expand our study to approximate versions of memorization—our work represents a strict lower bound on memorization which we have shown is already very high.
>
> > it's still hard for me to have a clear idea what types of sequences are more easily extracted
>
> We can include more examples of memorization in the appendix of our paper to show this. A study of what kinds of text are more memorizable than others is an important subject for future work.
>
> >  I find this paper lacking guidance or insights on how we can reduce repetition
>
> The goal of our paper is to study the phenomenon rigorously. Once we have shown the effect is true, future work can aim to solve the problem. However the first step is to understand the effect, and that is our goal here.

---

### Official Review · Reviewer_G66g · 2022-10-24

**Confidence:** 4
**Correctness:** 4
**Technical Novelty And Significance:** 3
**Empirical Novelty And Significance:** 3
**Recommendation:** 8

**Clarity, Quality, Novelty And Reproducibility:**

Some comments:

Intro, paragraph 3: “While McCoy et al. (2021) broadly study the extent to which language models memorize, their focus is on how to avoid the problem and ensure novelty of model outputs, rather than on studying model risk through identifying the maximal amount of data memorization.” - I believe this description of the work by McCoy is not accurate. They focus on evaluating novelty rather than ensuring it.

Typos:

Abstract: “models continues”

Section 3.2, paragraph 1: “throughput”

**Strength And Weaknesses:**

Strengths:

1) The problem is important

2) The methodology is solid, the experiments are thorough

3) The definition of memorization is reasonable and makes a lot of sense when dealing with language models: the paper considers generated examples and not e.g. token-level accuracy or loss

Weaknesses:

The results are expected and some of the trends in one way or another were reported in previous work.
However, I believe that overall the paper gives a solid ground for future work on memorization.


**Summary Of The Paper:**

The paper studies memorization in large language models and tries to quantify memorization rates depending on prefix length, duplication rate in training data and model size. The authors define an example memorised (given prefix of some length) if greedy decoding conditioned on this prefix reproduces the training example verbatim. This allows to estimate memorization rate in a more thorough manner differently from e.g. extraction attacks done in previous work. Experiments are conducted for different datasets (Pile, C4), models (GPT, T5, etc) and settings (e.g., with or without deduplication). The authors find that memorization rate is log-linear with respect to model size, prefix length and duplication rate in training.

**Summary Of The Review:**

The paper measures memorization for NLMs by looking at how often output of greedy decoding given prefix reproduces training examples verbatim. The experiments show log-linear memorization rate with respect to model size, prefix length and duplication rate in training. The results are not surprising, but given solid methodology and thorough experiments I believe overall the paper is of good quality and can give a solid ground for future work on memorisation.

---

> ### Author Response · Authors · 2022-11-18
> **Author Response**
>
> We agree with the reviewer that several of the trends we report are expected given anecdotes provided in prior work; the goal of our paper is to make these collections of beliefs something that can be actually measured empirically for future work to use with confidence. We are the first to show models memorize orders of magnitude more data than previously believed.
>
> We will update the paper to address the clarity suggestions, and revise our description of McCoy et al.

---

### Official Review · Reviewer_vuN8 · 2022-10-24

**Confidence:** 4
**Correctness:** 4
**Technical Novelty And Significance:** 2
**Empirical Novelty And Significance:** 2
**Recommendation:** 6

**Clarity, Quality, Novelty And Reproducibility:**

The paper is clear and reproducible.
It has limited novelty.

**Strength And Weaknesses:**

Strength:It's an interesting and intuitive idea in the paperI found the paper to be easy to read and understand. Very interesting results and analysis.

Weaknesses:
- Limited novelty

**Summary Of The Paper:**

This paper proposes a simple method to measure how well LLMs memorize training data. The authors found that memorization increases when 1) the model size is increased 2) the sample size is increased 3) the context of the example is increased.
Considering the level of memorization of T5 and comparing it with causal language models of similar size is the most interesting part of this paper for me.

**Summary Of The Review:**

This is an important topic, and the paper is interesting. Although the idea is not that novel, I think this is a good addition to the conference.

---

> ### Author Response · Authors · 2022-11-18
> **Author Response**
>
> We appreciate the reviewer’s comments on the fact that this paper has interesting results and analysis. We would be curious to hear what “limited novelty” concerns the reviewer has – to the best of our knowledge, we are the first to rigorously study how memorization scales with model size or training data repetition, and we show that prior estimates of the fraction of content memorized by a language model under-count by over 1,000,000x.

---

### Decision · Program_Chairs · 2023-01-20

**Decision:**

Accept: notable-top-25%

**Justification For Why Not Higher Score:**

The paper's findings are of wide interest due to the current interest in the performance and behaviour of large language models. However, the paper does not have enough novelty to justify an oral presentation.

**Justification For Why Not Lower Score:**

The spotlight recommendation is mostly based on the current wide interest in the performance and behaviour of large language models, which means that the paper’s findings might be of interest to a larger audience. Otherwise it would also be suited as a poster presentation.

**Metareview: Summary, Strengths And Weaknesses:**

The paper performs an extensive study of memorization in large neural language models. It proposes a methodology to quantify memorization rates and perform experiments on a number of language models. A number of factors influencing memorization are identified, including model capacity, data duplication and prompt length). According to the paper’s methodology, a much larger proportion of training data is memorized by large language models such as GPT-J than claimed by previous work. While memorization in language models has been studied before, and most of the findings are not surprising, the paper does provide more extensive empirical evidence of memorization than previous work.

**Note From Pc:**

if the above contains the word "oral" or "spotlight" please see: "oral" presentation means -> notable-top-5% and "spotlight" means -> notable-top-25%. As stated in our emails, we are disassociating presentation type from AC recommendations